# “Do Elite Sport First, Get Your Period Back Later.” Are Barriers to Communication Hindering Female Athletes?

**DOI:** 10.3390/ijerph182212075

**Published:** 2021-11-17

**Authors:** Martina Höök, Max Bergström, Stig Arve Sæther, Kerry McGawley

**Affiliations:** 1Swedish Winter Sports Research Centre, Department of Health Sciences, Mid Sweden University, 831 25 Östersund, Sweden; martina.hook@miun.se (M.H.); max.bergstrom@miun.se (M.B.); 2Swedish Ski Association, 791 31 Falun, Sweden; 3Department of Sociology and Political Science, Norwegian University of Science and Technology, NTNU, Dragvoll, 7491 Trondheim, Norway; stigarve@ntnu.no

**Keywords:** coach–athlete relationship, communication, focus group, interview, menstruation, sport, women

## Abstract

Competitive female athletes perceive their hormonal cycles to affect their training, competition performance and overall well-being. Despite this, athletes rarely discuss hormonal-cycle-related issues with others. The aim of this study was to gain an in-depth understanding of the perceptions and experiences of endurance athletes and their coaches in relation to barriers to athlete–coach communication about female hormonal cycles. Thirteen Swedish national-/international-level female cross-country skiers (age 25.8 ± 3.6 y) and eight of their coaches (two women and six men; age 47.8 ± 7.5 y) completed an online survey relating to their educational background, prior knowledge about female hormonal cycles and a coach–athlete relationship questionnaire (CART-Q). They then participated in an online education session about female hormonal cycles and athletic performance before participating in semi-structured focus-group interviews. Thematic analyses revealed three main barriers to communication: knowledge, interpersonal, and structural. In addition, the results suggested that a good coach–athlete relationship may facilitate open communication about female hormonal cycles, while low levels of knowledge may hinder communication. To overcome the perceived barriers to communication, a model is proposed to improve knowledge, develop interpersonal relationships and strengthen structural systems through educational exchanges and forums for open discussion.

## 1. Introduction

Research suggests that a strong coach–athlete relationship is an important factor for success in elite sports [1]. Coach–athlete relationships are based on communication (i.e., the sharing of information) and trust in pursuit of a common goal [1,2,3]. Jowett and Ntoumanis [4] identified three interpersonal constructs that describe the coach–athlete relationship, which have been termed the 3Cs: commitment (interpersonal thoughts; motivation to maintain a close relationship over time), closeness (interpersonal feelings; mutual respect, trust, appreciation and liking for one another) and complementarity (interpersonal behaviors; leadership and co-ordination). Previous research has shown that athletes participating in individual sports experience greater levels of commitment, closeness and complementarity with their coaches compared with team-sport athletes [5]. Furthermore, Lorimer and Jowett [6] observed a better empathic understanding among athletes and coaches from individual versus team sports. The same authors also showed that having an empathic coach might have a positive impact on an athlete’s performance and success, implying a need for coaches and athletes to work closely together [7]. The coach–athlete relationship may be strengthened, with the athlete perceiving the coach as more reliable and trustworthy [1] if effective communication strategies are developed [8]. This could include discussing issues relating to female hormonal cycles, a topic that is often overlooked within applied sporting environments [9,10], likely due to a sense of discomfort, particularly among male coaches [11,12].

Female athletes experience a variety of physical and emotional symptoms as a result of their menstrual cycle (MC), which may negatively affect their health, well-being and athletic performance [13,14,15]. In addition, negative side effects have been reported with female hormonal contraceptive (HC) use, including depression [16], reduced maximal aerobic capacity [17], inferior responses to sprint training [18] and weight gain [19]. Despite these challenging circumstances, athlete–coach communication about female hormonal cycles, including the MC and HC use, is limited [14,15,20]. For example, Solli et al. [15] reported that only 27% of 140 elite female endurance athletes had discussed the MC with their coaches. Notably, this percentage was lower (22%) if the athletes had a male coach and higher (44%) if the athletes had a female coach. Brown et al. [20] and Findlay et al. [14] have presented similar findings, with female athletes from both individual and team sports reporting more difficulty in discussing their MC with male coaches. The general lack of communication on this topic may be due to athletes perceiving their MC as private, taboo, uncomfortable, awkward and embarrassing, as well as a perception of having insufficient knowledge [15,20].

Limited knowledge about the effects of the MC and HC use on health, well-being and athletic performance has been reported by female athletes [15,21] and their coaches [11]. Solli et al. [15] reported that only 8% of elite female endurance athletes felt that they had sufficient knowledge about female hormonal cycles, despite the majority wanting to know more about how the MC and HC use affect training, physical adaptations and performance. The athletes also felt that their coaches (81% of whom were men) lacked knowledge in this area. Low levels of knowledge may be one explanation for athletes rarely planning or modifying their training routines according to their MC, which was also reported by the authors. Larsen et al. [21] also described a low level of knowledge about the MC and HC use in a sample of 189 female athletes. These authors reported higher knowledge scores among athletes competing in individual sports compared with team sports, and higher knowledge scores in HC users compared with non-HC users. However, the magnitude of this difference was low, and overall knowledge was generally considered to be poor. To overcome the issue of insufficient knowledge, it has been proposed that more attention ought to be paid to educating female athletes and their support teams about the female MC and HC in relation to training, as well as facilitating positive conversations and building the confidence to talk about the MC [20].

In summary, numerous recent studies have highlighted a substantial limitation in communication about the MC and HC use among athletes and their coaches, which may lead to negative effects on female athlete health, well-being and performance. This phenomenon appears to be at least partly due to a variety of inter-related factors, such as the strength of the coach–athlete relationship, feelings of social discomfort, the gender of the coach and a lack of knowledge about the subject area. The aim of the present study was to gain an in-depth understanding of the perceptions and experiences of elite female endurance athletes and their coaches in relation to barriers to communication about MC and HC issues. Based on previous recommendations [14,15,20,21], and due to the applied nature of the present study, we also aimed to increase knowledge and communication among the participating athletes and coaches through the research process.

## 2. Materials and Methods

### 2.1. Participants

The present study included 13 elite female cross-country skiers from the Swedish national team (age 25.8 ± 3.6 y) and 8 coaches who were working as national team coaches or personal coaches to the participating athletes (two women, six men; age 47.8 ± 7.5 y). The total number of participants was considered sufficient for exploring rich data, also described by Malterud et al. [22] as information power. The study was carried out according to the Declaration of Helsinki and was pre-approved by the Swedish Ethical Review Board (reference 2021-01047). The participants were informed that their participation was voluntary and that they could withdraw from the study at any time during the research process until the article was published. They also received all necessary information about the study objectives before providing written consent. To ensure confidentiality, the participants were given pseudonyms, and only the authors had access to their personal information and the collected data.

### 2.2. Procedures

#### 2.2.1. Overview

The data collection was completed in four sequential stages (Figure 1): (1) an online survey; (2) an online education session; (3) small focus-group interviews with athletes and coaches in separate groups; (4) small focus-group interviews with athletes and coaches together.

#### 2.2.2. Online Survey

The online survey (stage 1) was organized into three sub-stages: (1a) educational background; (1b) prior knowledge about female hormonal cycles; (1c) an 11-item coach–athlete relationship questionnaire (CART-Q) [4]. Educational background was based on the participants’ highest level of education, ranging from upper secondary school to university, and the types of university courses studied (Table 1 and Table 2). Prior knowledge about female hormonal cycles was assessed based on questions used by Solli et al. [15] and Larsen et al. [21]. The participants were asked how many times they had participated in educational sessions (e.g., presentations delivered as part of a university course or internally by the Swedish Ski Association) about the female hormonal cycle prior to the study (Table 1 and Table 2). The athletes were also asked whether they had spoken to a coach about their hormonal cycles at least once during the previous year and whether they wanted to know more about the MC and HC in relation to training, performance and development. Similarly, the coaches were asked whether they had spoken to their athletes about these topics in relation to training, performance and development. To assess the coach–athlete relationships, we used the validated Swedish version of the athlete CART-Q [23] and a translated Swedish version of the coach CART-Q. Briefly, the questionnaire contains 11 items measuring commitment (3 items), closeness (4 items) and complementarity (4 items). Based on a 7-point scale ranging from 1 (strongly disagree) to 7 (strongly agree), the participants rated how they perceived the quality of their relationship with their athlete/coach in response to statements such as: “I feel committed to my athlete/coach” (commitment), “I like my coach” (closeness) and “I am ready to do my best for my coach” (complementarity). Mean (SD) CART-Q scores are presented in Table 1 and Table 2 with a higher score (e.g., closer to 7) representing a higher degree of commitment, closeness and/or complementarity.

#### 2.2.3. Online Education Session

All participants took part in a theoretical lecture (stage 2), which was delivered online (due to COVID-19 restrictions) by a professor of gynecology and obstetrics working closely with elite athletes. This education session lasted approximately 60 min and included basic knowledge about the MC, HC use and symptoms in relation to athletic performance. There was also approximately 20 min after the lecture for questions and answers. The purpose of stage 2 was to provide the participants with a baseline understanding of fundamental female hormonal physiology (e.g., what is considered a regular menstrual cycle, how much blood constitutes a heavy bleed, etc.), to enable questions around this topic to be answered accurately by all participants during the focus-group interviews (i.e., stages 3 and 4).

#### 2.2.4. Focus-Group Interviews

The first focus-group interviews (stage 3) enabled the participants to initially discuss their experiences with their teammates (athletes) or colleagues (coaches), while for the second interviews (stage 4) athletes and coaches were grouped together. In stage 3, the groups contained 3–5 athletes or 3–4 coaches. In stage 4, which took place approximately three weeks after stage 3, the number of participants varied between 3 and 5 and included 2–4 athletes and 1–2 coaches, which is consistent with previous recommendations [24,25]. For these mixed interviews, the athletes were grouped together with at least one coach with whom they worked closely (e.g., their national team or personal coach).

The focus-group interviews were organized as discussions, in order to provide a greater depth of interaction between participants [24]. In addition, the interviewer ensured that all participants had an opportunity to contribute. The interview guide was inspired by and developed from previous studies on the female hormonal cycle [13,15]. The interviews focused mainly on gaining an overall insight into the coach–athlete relationship in relation to the female hormonal cycle and the participants’ experiences of and knowledge about the MC and HC use, including how the female hormonal cycle impacts athlete performance. Consistent with Gratton and Jones [24], the interviewer took the role of facilitator, keeping the discussions relevant to the study aim by using probes or asking open questions such as “How do you feel about discussing issues concerning the menstrual cycle and hormonal contraception?” or “Has your approach to dealing with issues such as the menstrual cycle, hormonal contraception, symptoms and performance changed during your athletic career (athletes only)/the time you have worked as a coach (coaches only)?”. The interviews were led by the first author, who possessed a well-developed contextual understanding of the group through working as a coach within the Swedish Ski Association. Therefore, a trustful relationship with the participants had been established prior to the study, which was believed to generate richer interview data. All interviews were recorded in Microsoft Teams and lasted between 45 and 90 min.

### 2.3. Data Analysis

Mean and standard deviation (SD) descriptive statistics were calculated in Microsoft Excel for the 3Cs within the athlete and coach groups, separately. Coach–athlete relationships assessed by the CART-Q were considered of good quality for scores ≥5. The focus-group interviews were conducted in Swedish, the first language of all the participants, and translations to English for the purpose of international publication were agreed upon by at least two of the bilingual authors. The data were analyzed using thematic analysis, which is a process organized in the six following steps: (1) familiarizing yourself with the data, (2) generating initial codes, (3) searching for themes, (4) reviewing themes, (5) defining and naming themes, and (6) producing the report [26]. Two of the authors worked through steps 1–5 in parallel before comparing their findings. Initially, after transcribing the focus-group interviews, these two authors read through the text with an open mind, to get a general sense of the content. Interesting features from the entire dataset were then bunched into codes (step 2), such as communication, knowledge, and hormonal contraceptives, and clusters of sub-themes (step 3), such as limited knowledge (e.g., not knowing what is right or wrong). Initial sub-themes were developed and reviewed in step 4 of the analysis and were thereafter analyzed, refined and labelled into three main themes of barriers to communication (step 5). Finally, a number of quotes reflecting the themes in relation to the study aim and previous research were selected (step 6). Throughout the data analysis process the authors discussed various perspectives and interpretations of the themes to ensure peer validity [25,27].

## 3. Results

### 3.1. Online Survey

All participants except one of the athletes (Jennifer) had participated in at least one education session about the MC and HC use prior to the study. Most participants (10 athletes and 3 coaches) had participated in 1–2 education sessions throughout their careers, while some (2 athletes and 4 coaches) had been involved in three or more education sessions. Twelve of the thirteen female athletes in the present study stated that they had spoken to a coach about the MC or HC at least once during the previous year. All participating athletes (13) and coaches (8) reported that they wanted to know more about the female hormonal cycle in relation to training, competition and performance development. From the CART-Q questionnaire, the mean values for commitment, closeness and complementarity for athletes/coaches were 5.9/6.1, 6.8/6.8 and 6.5/6.6 out of 7, respectively (i.e., they ranked the quality of their coach–athlete relationship as good).

### 3.2. Focus-Group Interviews

The thematic analysis revealed three higher-order themes of barriers to communication about the female hormonal cycle between the athletes and their coaches: knowledge, interpersonal and structural (Table 3).

#### 3.2.1. Theme 1: Knowledge Barriers

Despite the athletes rating the quality of their coach–athlete relationships highly, with CART-Q scores consistently greater than 5 out 7 (see Table 1 and Table 2), several athletes perceived barriers for discussing the MC and HC use with their coaches. Moreover, while almost all of the athletes (12/13) had spoken to a coach about these topics in the previous year, they expressed feeling insecure about what was right or wrong (e.g., hormone levels, contraceptives, MC irregularities, etc.), and this knowledge barrier limited the level of communication with their coaches and teammates. This was exemplified by two athletes:

“I think, because you don’t have so much knowledge about it [the female hormonal cycle], then… I feel that when I start to talk about it I don’t really know what’s right or wrong. Or maybe there is no right or wrong. But it might be that you don’t really know so much about it.”(Kristina)

“There is some kind of uncertainty and you don’t really know [what’s right or wrong], so it’s hard to share that much with others when you don’t really know.”(Gunilla)

Several of the athletes also expressed doubt in the survey as to whether their coaches would be able to provide them with relevant answers to their questions concerning the MC and HC. This reduced the athletes’ efforts to initiate discussions relating to the topics, as mentioned by Josefin:

“So, contraceptives have sort of more functions than how I think men [male coaches] see contraceptives. Like, contraception isn’t only to stop us from getting pregnant, as many men think, I think. But contraception also has the ability to reduce hormonal fluctuations, it has the ability to reduce period pains and you end up maybe more in balance. So it’s not necessarily just in terms of protecting yourself against becoming pregnant. So I think it can be important to talk about it regardless [of why we use contraceptives]. Yeah, but like… who should take it up, I don’t know.”

Indeed, several coaches felt insecure about discussing the topic of HC use as they felt they lacked sufficient knowledge, as described by Arild:

“I think I… To talk about contraceptives for me, that feels a bit tough, in general.”

The participants were eager to optimize every detail of training and several of the athletes and coaches had considered implementing training according to the MC. However, the coaches perceived the MC as difficult to grasp, which led them to prioritize other factors. Interestingly, many of the athletes believed that MC issues did not concern them because they used the HC and did not experience a regular MC. As Gunilla explains:

“If you take the pill it’s like: “No, that [training according to the MC] doesn’t affect me anyway”, so instead you can hardly be bothered to listen… and maybe that’s stupid.”

Others had experienced an irregular MC but believed that this was normal or expected within elite endurance sport:

“I remember when I was a junior I heard this thing about menstruation. Yeah, kind of like, if you have your period then you’re not training enough.”(Celine)

“I have experience of that from when I went to the ski high school and was at that age that when I missed my period and, like, told my parents and maybe my ski high school coaches, it was more: ‘That’s completely normal… that’s what happens when you train a lot. Then it [your period] will disappear.’ And then I didn’t think there was anything wrong with me.”(Lotta)

Some of the coaches tried to stay up to date with research relating to the MC and HC, yet many experienced difficulties applying the research to the context of elite sports and sport-specific practice (in this case, cross-country skiing). Further, the coaches found individual needs and variations among the female athletes to be complex and difficult to make sense of. In an ideal world, they would have a clear framework to help them in their coaching roles (see Theme 3: Structural barriers, below). Some of the coaches had experience implementing MC-related research into the training plans of their female athletes, but felt that this practice had neither shown clear health benefits nor performance enhancements:

“I have a practical example [an athlete], in fact, who has done that actually since May, so planned according to it [the menstrual cycle]. And I can say that I have become more… even more questions now, half a year later, than I had back in the spring. There are no regularities in… there are regularities in menstruation, but not in performance.”(Samuel)

All of the coaches mentioned a frustration due to low levels of knowledge, as well as a lack of clear evidence for how to handle the MC in an optimal way. The following quote exemplifies the frustration expressed by several of the coaches:

“It’s damn hard! There are so many factors. We can’t standardize and say that this is because of the MC… there are so many parameters that affect form, so it’s hard.”(Oskar)

Instead of asking their coaches for advice, most athletes turned to doctors, nurses or other specialists (e.g., through the public health care system or medical experts/resources specifically connected to the national team) for support. However, the advice was often perceived as vague or too general for the athletes, making it hard to implement within an elite sport environment. The fact that the athletes generally consulted specialists seemed to confirm the coaches’ perceptions that discussion about the MC and HC use were outside of their coaching role and area of expertise. The coaches felt excluded from the conversations or did not know how to help, as expressed by Oskar:

“We shouldn’t barge into this area. Yeah, but [if an athlete asks]: ‘Shall we try and change contraception methods?’ ‘Yeah, I think so!’ I can’t stand there and say that! Well I can, but it wouldn’t be good!” [The group laughs]

#### 3.2.2. Theme 2: Interpersonal Barriers

In the survey, the athletes and coaches reported no or few perceived problems in talking about female physiology. However, several interpersonal barriers for discussing hormonal-cycle matters were expressed in the focus-group interviews, and it was identified by several participants that “we don’t do it so often”. Firstly, many athletes felt that they did not have any, or large enough, MC- or HC-related problems that warranted asking their coaches for advice. They also did not feel that hormonal cycles affected their health or sporting performance negatively, and therefore felt no need to discuss the subject with their coaches. Further, when reflecting on their sporting careers, the athletes had not seen any clear patterns in their performance that would require adaptations to their training in relation to their MC:

“If I’d had bigger problems, and felt that it affected me noticeably, then I might have mentioned it [my MC]. Because I think it’s important to perform well like, but I don’t think that’s a problem … No, so I don’t talk about it…”(Helena, who rated closeness to her coach as 7.0)

A common struggle among the athletes was highlighted in finding the best type of HC, and some expressed concerns about potential negative health effects as a result of long-term HC use. However, the topic of HC use was perceived as private and therefore difficult to discuss with others (e.g., teammates and coaches), even if the athletes stated a close relation with their coach, as expressed by Jennifer (who also rated closeness to her coach as 7.0):

“Of course it feels private, but it also feels quite important. I’ve sometimes thought, like, is it [contraceptive use] dangerous? Or will it harm my performance if I take this [contraception]? Like, you think about it and you kind of want to know, can I take this contraception without it negatively affecting me in some way? Because it’s kind of like… It still affects the body, so it’s more like that. You don’t know that much. You want to know more about it, like…”

Several of the athletes mentioned that it is easier to talk about MC and HC issues with their coaches now that they are older, compared to when they were junior athletes. However, while Kristina has experienced an increased willingness to talk about the topic as she’s matured, she still experiences some interpersonal barriers:

“Yeah, in the beginning, the first years, I could say to the coach that I had a stomach ache and I could go home, like. I mean, I didn’t even dare to say that I was on my period. And then I came back the next day, like. But, people have always been a bit like that. At least I have.”

Alexandra describes her experience as follows:

“I think it’s good to take this up as a junior too… Now I can be open and honest about it [menstruation], but I couldn’t have been like that as a junior. I couldn’t have talked about menstruation then… I didn’t have a single thought about talking to coaches about it, because I wouldn’t have dared.”

She continues with regard to the gender of her coach:

“But I think it would have been easier to talk about if it had been a female coach, for sure. Because that feels natural, like.”

Kristina also shared some thoughts about the coach’s gender:

“I don’t know if it would be different if I had a female coach. If that makes a difference too. Or that it kind of feels more… that they understand more. Even though… I don’t think it feels embarrassing to talk about the menstrual cycle. But just the understanding, that it might be easier to talk to a female coach.”

While being a female coach seemed to make it less difficult to talk about the MC and HC, it did not automatically provide the female coaches with a full understanding and the knowledge they felt they needed on the subject, as expressed by Erika:

“We have our own experiences, but it’s our body and no menstruation is the same. So just because I’ve had it one way doesn’t mean that all of my athletes experience it in the same way I have, that’s how it is… But then I know what period pains are and I know what it’s like to lie in bed in the fetal position for 24 hours, I know what that’s like… but that’s so different for everyone, so it’s really important even for us female coaches to broaden our knowledge and understand…”

Many of the male coaches reported avoiding discussions with their athletes about the HC, out of respect for their privacy and not wanting to encroach upon their sex lives. Several of the male coaches experienced the topic as “too private”, and having no lived experience of the MC or HC use seemed to add an extra barrier for the male coaches. This placed the topic outside of their comfort zone, despite stating high levels of closeness with their athletes (the mean closeness value was 6.8 for the male coaches). Moreover, it was not until a close coach–athlete relationship had been developed that the coaches felt they could really discuss both MC- and HC-related issues, as expressed by Arild.

“Us guys are a bit more out of our depth there. It’s a bit… there’s an extra barrier, so you need even more trust, I think, to be able to talk about this.”

As expressed by Daniel in the next quote, there is a degree of fear of crossing a line or being questioned by athletes as to why a coach would want to know about an athlete’s HC use, which he believed could damage the coach–athlete relationship (e.g., closeness and trust). This made him hesitant to take the initiative to start conversations:

“It can be difficult knowing whether the athlete you are meeting will feel the same… “Why should I share this with you?”, like: “Why do you want to know about that?”, or like: “Why is that relevant?”, and so on… And somehow it’s also quite private.”

Interestingly, there were no clear differences between the female and male coaches in how they experienced the 3Cs in their coach–athlete relationships (commitment: 5.8 and 6.2; closeness: 6.8 and 6.8; complementarity: 6.6 and 6.6). Nevertheless, the female coaches felt more confident in talking to athletes about the MC. For example, in the next quotation, Erika used terms like “natural” and “standard”. At the same time, she was thankful that none of her athletes had experienced any problems, which indicates that she may have felt insecure about how she would have responded had problems arisen, perhaps due to knowledge barriers:

“When I have training talks with my athletes a natural question is: “How is your training going?”, “How are you feeling about this [your MC]?”. And thank goodness, I’ve hardly had any athletes with a problem … I’ve never had anyone with problems… I always have it [the MC] as a standard question in all meetings with my [female] athletes. I’ve never had any problems bringing it [the MC] up.”

It appears that once a close coach–athlete relationship had been established, the coaches felt more confident in taking the initiative to start conversations. For example, Oskar (closeness 6.8) comments:

“I wouldn’t say it’s so bad that no-one talks about it at all. But it [talking about the MC] does require that you know them [the athletes] pretty well.”

#### 3.2.3. Theme 3: Structural Barriers

In general, the participants expressed insecurity about who should take the initiative to start MC-related conversations (e.g., how and when, and what to talk about). Even though spontaneous or informal discussions seemed to be inhibited by the previously described knowledge and interpersonal barriers, there were not many formal or organized forums in place within the sports association or national team. For example, most of the athletes and their coaches had only participated in 1–2 educational sessions or MC-related group discussions prior to this study (see Table 1 and Table 2). Rather, other competing interests (e.g., training, sleep, nutrition, physiotherapy/massage, logistics, etc.) were prioritized over hormonal-cycle matters (e.g., education and knowledge application).

There was no evidence of multi-disciplinary teams (e.g., coaches, medics, nutritionists, psychologists, etc.) working closely together to deal with female athletes’ hormonal cycles or of organized discussions or education sessions about the female hormonal cycle. This lack of structure may be due to other competing interests (e.g., prioritizing sporting results) and/or limited knowledge (see Theme 1: Knowledge barriers) and may have a negative impact on athlete health, well-being and performance. One coach, Arild, found this lack of structure (i.e., limited communication between medical doctors, nutritionists and coaches) to be a problem if athletes used the HC to mask other issues:

“So I think they [the athletes] play along. The ones who have some idea about this, they know that they can say that they don’t have their period because they take the pill and that’s true in a way, but really it’s because they are too thin and that’s damn hard to deal with if you [the coach] can’t follow up on this [the athlete’s health] whenever needed.”

Tommy also describes a perceived dilemma for coaches and athletes due to limited communication between medical doctors, nutritionists and coaches:

“To get a regular period we might have to increase the body weight and then there will be a clash for the athlete… “gain weight, [but] I need to compete for some more years, so what do I do?” And then you have created a bigger problem in their minds, so it isn’t easy.”

For many of the coaches it felt as if the athletes had to choose between having a normal MC (i.e., eumenorrhea) and prioritizing their sporting performance. From their experience of dealing with recommendations from nutritionists (i.e., to increase carbohydrate content in the diet) and medical experts (i.e., to increase fat content in the diet), becoming eumenorrheic meant gaining weight and reducing training volume, which could negatively affect competitive performance. Hence, in the coaches’ eyes, based on the expert recommendations (i.e., “eat more fat”), an irregular MC (i.e., oligomenorrhea or amenorrhea) was something the athletes might have to accept if they want to perform at an international level. Oskar described this as follows when he discussed it with the athletes:

“The best thing for your body is to not do elite sport, as you do. The best thing in the world, that’s to have some regularity [in your menstrual cycles], yes, but: ‘Now I’ll eat more fat to get my period back’… that doesn’t go hand-in-hand with elite sport.”

This was also expressed by one of the athletes, Amanda:

“When I started to discuss this with the coaches it was more like: ‘Yeah, maybe you should do one thing at a time; do elite sport first and get your period back later. Maybe you can’t expect to have both at the same time…’ And that felt a bit like: That’s not how I want it to be, I feel like it should be possible to have periods and optimize [my] performance.”

The impact of the MC and/or HC use generally seemed to be considered by athletes and coaches as somewhat unknown or hard to measure, in comparison to other factors that affect cross-country skiing performance (such as lactate threshold, VO_2_max, energetics, technique, strength, etc.). This abstractness contributed to the issue of communicating about the topic, and while other variables were measured regularly, there never seemed to be a right time to implement potential changes into the training program in relation to the MC, as described by Elisabet:

“Based on what a year of training looks like it’s quite difficult to find the time to make changes, because you can’t do it during the season and it wouldn’t add so much doing it in April during the recovery period where the training volume is lower.”

Amanda also expressed difficulties when trying to implement training according to her MC:

“It sounds exciting when you hear of studies about how strength training and stuff can have an effect… and then you had hoped that you could include it in your own training plan. I took it up with Oskar and Tommy, that: Now I’ve stopped taking the pill, so I’ll let you know when I get my period back [so that I could train according to my menstrual cycle]. And then half a year later: No, it’s not time yet [I still haven’t got my period back].”

Although most athletes had questions and experiences related to female hormonal cycles and elite sport, they seldom discussed these issues with their teammates. Many felt uncomfortable and insecure regarding how and when to instigate discussions and felt that the issues were outside of their normal conversational topics, as expressed by Kristina:

“Yeah, but I don’t feel very comfortable with it. It’s probably not, like, me who brings it up in the first place either.”

Interestingly, as expressed by Amanda in the next example, when the female athletes had shared their experiences with each other the majority could relate to one another.

“I started taking the pill when I was maybe 15, and it was the nurse back home in [hometown] or [another town]… and since then it’s just carried on. I haven’t, like, asked anyone else what they do. So that was an eye-opener for me [discussing the female hormonal cycle with other teammates in an organized way].”

Bodil and Lena suggested that it would have been beneficial for them to have been exposed to a similar educational intervention (as in this study) earlier, as junior athletes. They also mentioned that they probably would have started to share experiences with other athletes and coaches earlier in their careers if they had had a better understanding and knowledge of the female hormonal topic as juniors. Several other athletes had similar positive experiences once someone else had opened up the conversation, for example, after participating in the online lecture together. However, such occasions were described as quite seldom, due to the lack of formal structure for conversations. The coaches also described female hormone-related discussions with athletes as relatively unusual and in some cases perceived the silence as an unwillingness to raise related questions:

“I would say that it’s very, very rare, if it even happens at all, that the girls take it up. It’s more like the opposite, that you have to ask.”(Hampus)

## 4. Discussion

The aim of the present study was to provide an in-depth understanding of the perceptions and experiences of athletes and coaches in relation to communication about MC and HC use. The study also aimed to increase knowledge and communication among the participating athletes and coaches through the research process. In contrast to previously published studies, our study focused on both the athletes’ and coaches’ perspectives of the same phenomenon and provided an opportunity for interaction between the two groups.

Previous studies have reported MC-related symptoms to be common among elite endurance athletes [13,14,15]. While we did not collect systematic questionnaire data on athletes’ symptoms in the present study, during the interviews the athletes seldom referred to symptoms or issues related to their MC or HC use that they felt were worth discussing with their coaches. This is similar to the findings of Brown et al. [20], where a number of athletes reported not talking about their MC with their coaches because they were unaware of the potential impact that the MC could have on performance. It could be speculated that a perceived lack of symptoms or issues may have helped the athletes in the present study to reach an elite level in their sport. An alternative explanation is that their high performance level has provided them with access to advanced medical expertise. Also worth noting is that several of the athletes using the HC believed that MC matters did not concern them. While this may be true for some women, others can be unconsciously affected by HC use [21]. Moreover, HC use can mask irregular or unhealthy hormonal conditions in women [28]. Given the limited knowledge of the MC and HC use reported by the athletes and coaches in the present study, it is important that expert practitioners help to monitor the health, well-being and athletic performance of female athletes in relation to their individual needs.

When surveyed, the participants reported that they perceived no or only minor barriers to communicating about female hormonal topics, yet results from the focus-group interviews revealed several barriers to communication for the athletes and coaches. Knowledge was identified as one of three higher-order themes, which is supported by previous research demonstrating low levels of knowledge about the MC and HC use among Norwegian [15], Australian [21] and British [20] athletes. The reluctance of the athletes in the present study to initiate discussions may have been due to their limited knowledge of how their health, well-being and athletic performance could be affected by their female hormonal cycles. This notion has been reported in medical care, where women are often unaware of or have misconceptions about conditions that affect their sexual health, making them abstain from seeking help [29]. Additionally, many of the athletes in the present study believed that MC- and HC-related issues were outside the knowledge area of their coaches, which has been reported previously by elite cross-country skiers in Norway [15]. While all of the coaches in the present study had talked about the MC or HC use at least to some degree with their athletes, they did express feelings of insecurity and discomfort in these situations, particularly the male coaches. This is similar to previous findings showing male coaches to be less likely to talk to athletes about MC issues than female coaches [12], which may also be linked to the “unease” reported by female athletes when conversing with male coaches about the female hormonal cycle [14,15].

There were some differences in how the athletes ranked their closeness to their coach when completing the CART-Q, which could potentially affect how comfortable they felt about raising topics related to the MC and HC use. Kristina mentioned a degree of discomfort in talking about hormonal cycle issues, and her mean closeness rating was 6.3, which was slightly lower than the mean group score of 6.8. Whether this is a function of specific coach–athlete relationships or the athlete’s personality traits is unclear. For example, Kristina mentioned that she has always found it difficult to talk about her MC with coaches, ever since she was a junior. This is common for young women, who are often socialized into concealing their menstrual experiences [30]. In general, the athletes described a willingness to discuss the MC and HC use with their coaches and this could potentially be related to the quality of the coach–athlete relationships. Paradoxically though, some coaches appeared to avoid discussing HC use with their athletes for fear of damaging the coach–athlete relationship. Another factor related to the willingness of the participants to discuss the MC and HC use in the present study was a mutual dominant focus on performance optimization. This is consistent with the results of Brown et al. [20], who suggested that a performance-orientated focus might be beneficial when raising female-hormone-related topics.

While a lack of knowledge and interpersonal factors appears to form specific barriers to communicating about the female hormonal cycle, structural barriers were also identified in the present study as a higher-order theme. This is similar to findings from Clarke et al. [11], who suggested that communication about the MC may be inhibited if coaches are only able to rely on their personal experiences and interpersonal skills without the support of an evidence-informed framework. Consistent with several previous studies (e.g., [14,20]), we identified a need for structures and interventions that would facilitate discussions and the exchange of knowledge and experiences. A closer connection and transparency between athletes, coaches, medical experts, nutritionists and psychologists may help coaches to understand where their responsibility ends and where subject-specific experts should take the conversation further, which was identified in the present study as an issue. A similar concept has been suggested by Clarke et al. [11], who recommended the development of a female-athlete–coach education framework that is co-developed with coaches, athletes, sport scientists, dieticians, psychologists and medical staff. As a concrete example, the participants in the present study suggested that educational interventions relating to female hormonal cycles ought to be included in athlete development programs from a younger age (e.g., as juniors). It was highlighted that such interventions should include information relating to female-specific physiology and the broader effects of the female hormonal cycle on health, well-being and performance in the context of elite sports.

In general, the athletes appreciated sharing their thoughts and experiences with each other and were positive about the concept of the present study. Moreover, the coaches were able to increase their understanding of their athletes through open and guided conversations. Combining an educational lecture with guided focus-group discussions seemed to help all participants in overcoming at least some of the described barriers to communication. However, a number of concerning issues were raised by the participants, not least the belief that an irregular or absent MC is acceptable or perhaps even a requirement in elite sport. Beyond the scope of the present research, it is worth noting the important role that governing bodies play in sports where leanness is a key determinant of successful performance, as is the case in cross-country skiing [31]. For example, modifying cross-country skiing competition tracks to include fewer vertical meters (i.e., less climbing) may result in a reduced focus on body mass and thereby reduce the risk for female hormonal issues among elite athletes. The present study also highlights a need for structured educational and communication frameworks, as well as routines for regularly monitoring female athletes’ hormonal status in relation to health promotion and performance optimization. Viewing the female hormonal cycle as a natural performance variable in women, rather than an abstract and embarrassing occurrence, may help to promote positive attitudes about female athlete health and enhance women’s sporting performance.

### 4.1. Limitations

There are a number of limitations worth acknowledging in the present study. Firstly, the athletes were elite cross-country skiers representing the senior national team, and the coaches were from the senior national team and/or were personal coaches to the participating athletes. Therefore, the results may not be generalizable to younger athletes, lower-level performance groups and/or other sports. Another limitation may be observed in the Swedish version of the CART-Q questionnaire, which has been validated but might not be as accurate as expected. For example, we identified some issues with the translation of the question: “I am committed to my coach/athlete” in the commitment category, a problem that has been highlighted previously [23], and would potentially result in a lower mean commitment score than intended. Further, participation in the present study was voluntary, which may have led to a bias in the sample group. It is possible that the volunteer participants may have been more open-minded and interested in discussing female hormonal cycles than their non-participating peers. The gender distribution among the coaches may also have affected the opinions represented in the study, with more men (n = 6) versus women (n = 2). This was due to the gender split at this level in cross-country skiing, where the majority of the coaches are men. Finally, the interviewer in the present study worked part-time as a coach within the national ski association, which may have affected the participants’ openness either positively (due to well-developed relationships) or negatively (due to feelings of an obligation to participate and/or respond in a specific way).

### 4.2. Practical Applications and Future Research

To summarize our findings and guide future work we have developed a working model for practitioners and researchers (Figure 2). The model is a development of the methods used in the present study (see Figure 1) and includes suggestions for increasing knowledge and facilitating communication about female hormonal topics among athletes and their support teams (i.e., coaches, medics, nutritionists, psychologists, etc.), with respect to the barriers we identified (i.e., knowledge, interpersonal and structural). Since the athletes expressed a desire to have participated in a similar process earlier in their athletic careers, we recommend starting this intervention with junior athletes. In step 1, we propose measures of existing knowledge levels, coach–athlete relationship quality and communication pathways (e.g., structural forums/platforms) to assess the size and scope of the existing problem. This may be achieved by surveying athletes, coaches and other practitioners. An educational intervention is then recommended (step 2), which can provide the group with basic knowledge of female-specific physiology in the context of athletic performance. This education should be delivered by a specialist with relevant knowledge and experience. From here, focus-group discussions can be used as a method to gain a deeper understanding of athletes’, coaches’ and practitioners’ perceptions and experiences related to the MC and HC use (steps 3a and 3b). The final part of the model proposes a follow-up to develop future plans for supporting the long-term development of female athletes (step 4). As shown in Figure 2, the process can continuously cycle through steps 1–4 as athletes progress through their careers, thereby allowing the working group to evolve together.

## 5. Conclusions

In this study we used a survey, an educational session and two semi-structured focus-group interviews to investigate the perceptions and experiences of athletes and coaches in relation to barriers for athlete–coach communication about the female hormonal cycle. The main findings were that the 13 female athletes and 8 coaches (2 women, 6 men) experienced knowledge, interpersonal and structural barriers when discussing the female hormonal cycle. Perceived low levels of knowledge hindered communication between athletes and coaches. A strong coach–athlete relationship seems to be one factor that may reduce the barriers to communication. Organized forums are recommended to promote and facilitate regular communication. These forums may include educational interventions (starting when the athletes are at a junior level) and open conversations between athletes, coaches, medical experts, nutritionists and psychologists. The present study provides an example of how to increase open communication about female hormonal cycles among athletes and coaches. Future work should focus on implementing longitudinal interventions, starting earlier in athletes’ careers, to support the optimal development of women participating in sport.

## Figures and Tables

**Figure 1 ijerph-18-12075-f001:**
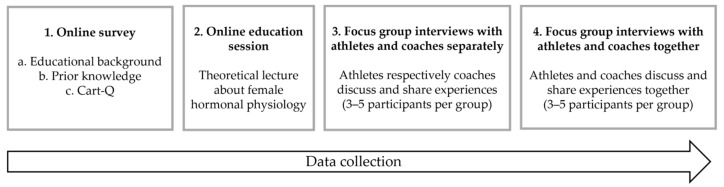
The four sequential stages of the data collection.

**Figure 2 ijerph-18-12075-f002:**
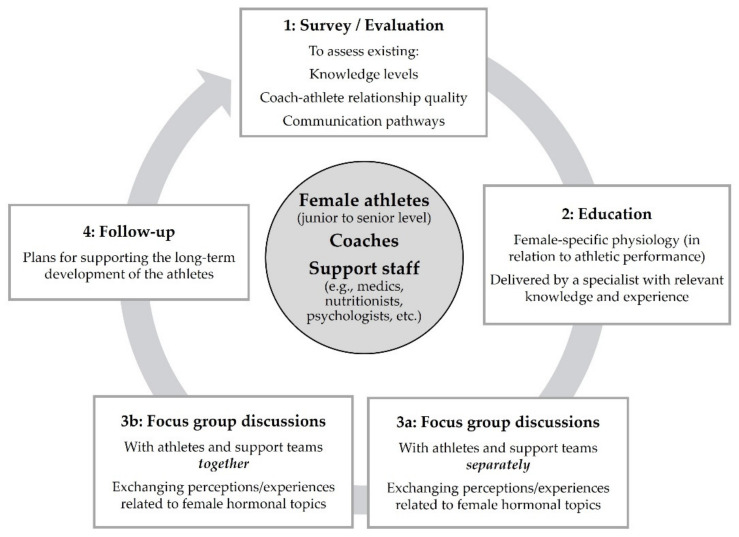
A working model for overcoming barriers to communicating about the female hormonal cycle.

**Table 1 ijerph-18-12075-t001:** Descriptive data for the female athletes relating to age, educational level, prior education * and mean (standard deviation, SD) coach–athlete relationship (CART-Q) scores.

Pseudonym	Age (y)	Highest Level of Education	Prior Education *	Commitment	CART-Q Closeness	Complementarity
Lena	20–25	Upper secondary school	3–4	6.0 (1.4)	7.0 (0.0)	6.3 (1.3)
Celine	20–25	University, other	1–2	5.3 (1.2)	6.8 (0.4)	6.3 (0.4)
Helena	20–25	University, other	1–2	6.7 (0.5)	7.0 (0.0)	7.0 (0.0)
Josefin	20–25	University, sport sci	1–2	6.0 (1.4)	7.0 (0.0)	7.0 (0.0)
Greta	20–25	University, other	1–2	5.0 (2.8)	7.0 (0.0)	7.0 (0.0)
Alexandra	20–25	University, sport sci	3–4	7.0 (0.0)	6.8 (0.4)	5.0 (1.2)
Lotta	20–25	University, other	1–2	5.0 (2.8)	7.0 (0.0)	7.0 (0.0)
Bodil	20–25	University, other	1–2	6.5 (0.5)	7.0 (0.0)	7.0 (0.0)
Jennifer	26–35	University, other	0	6.0 (1.4)	7.0 (0.0)	7.0 (0.0)
Kristina	26–35	University, other	1–2	6.0 (0.0)	6.3 (0.4)	5.3 (0.4)
Marie	26–35	University, other	1–2	5.7 (1.2)	7.0 (0.0)	6.8 (0.4)
Gunilla	26–35	Upper secondary school	1–2	5.0 (2.2)	7.0 (0.0)	7.0 (0.0)
Amanda	26–35	University, sport sci	1–2	5.7 (1.2)	5.5 (0.9)	6.0 (0.0)
			Mean (SD)	5.9 (0.6)	6.8 (0.4)	6.5 (0.7)

* Number of prior menstrual cycle/hormonal contraceptive educational sessions received. Sport sci: sports science courses; other: courses not containing sports science.

**Table 2 ijerph-18-12075-t002:** Descriptive data for the female (F) and male (M) coaches relating to age, educational level, prior education * and mean (standard deviation, SD) coach–athlete relationship (CART-Q) scores.

Pseudonym	Age (y)	Highest Level of Education	Prior Education *	Commitment	CART-Q Closeness	Complementarity
Elisabet (F)	30–45	University, sport sci	>5	6.0 (0.8)	6.5 (0.5)	6.5 (0.5)
Erika (F)	46–60	University, coach education	0	5.7 (1.2)	7.0 (0.0)	6.8 (0.4)
Daniel (M)	30–45	University, sport sci	>5	5.7 (1.2)	6.8 (0.4)	6.5 (0.5)
Samuel (M)	30–45	University, coach education	1–2	7.0 (0.0)	7.0 (0.0)	7.0 (0.0)
Tommy (M)	46–60	University, sport sci	1–2	7.0 (0.0)	7.0 (0.0)	6.5 (0.9)
Hampus (M)	46–60	University, coach education	3–4	6.3 (0.5)	6.5 (0.5)	6.8 (0.4)
Oskar (M)	46–60	University, other	1–2	6.3 (0.9)	6.8 (0.4)	6.3 (0.4)
Arild (M)	46–60	University, other	>5	5.0 (1.4)	7.0 (0.0)	6.8 (0.4)
			Mean (SD)	6.1 (0.6)	6.8 (0.2)	6.6 (0.2)

* Number of prior menstrual cycle/hormonal contraceptive educational sessions received. sport sci: sports science courses; other: courses not containing sports science/coach education.

**Table 3 ijerph-18-12075-t003:** Higher-order themes of barriers to communication related to the female hormonal cycle derived from the focus-group interviews with the athletes and coaches.

Main Theme	Athlete Perspective	Coach Perspective
Knowledge	Limited knowledge (e.g., not knowing what is right or wrong) Concerns expected to be outside the knowledge area of the coach (e.g., contraceptives) General MC recommendations perceived as vague in relation to the elite athlete context Believing that MC matters do not concern them, such as HC use	Lack of MC-/HC-specific knowledge and frameworks/guidelines Concerns considered to be outside their knowledge area (e.g., contraceptives) Challenging to incorporate MC research into sport-specific practice (e.g., training plans, individual variation and complexity of athlete performance—lack of useful guidelines) Doubts regarding potential benefits of training according to the MC (e.g., optimizing athlete performance is the highest priority, the MC is just one piece of the puzzle)
Interpersonal	Not having any, or large enough, MC-related problems Feeling uncomfortable/inexperienced discussing the MC and HC use with coaches or teammates (but more natural talking to a female coach) Coach–athlete relationship (e.g., closeness, level of trust and confidence between coach and athlete)	Perceived taboo and respecting athletes’ privacy Topic beyond the comfort zone or feeling unsure how to help athletes (e.g., timing and approach) Coach–athlete relationship (e.g., closeness, level of trust and confidence between coach and athlete)
Structural	Lack of formal or organized discussion forums/education No or little experience of teammates sharing their hormonal matters or questions with each other The endurance athlete dilemma (e.g., the importance of being light weight and training enough versus having a regular MC)	Lack of formal or organized discussion forums/education Lack of structures in place for discussing the female hormonal cycle with athletes, coaches and support staff (e.g., medical, nutritionist, etc.) The coach dilemma (e.g., the importance of athletes being light weight and training enough versus having a regular MC)

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
