# Peer review of "“Do Elite Sport First, Get Your Period Back Later.” Are Barriers to Communication Hindering Female Athletes?"

_ijerph, 2021, doi:10.3390/ijerph182212075_

Round 1

Reviewer 1 Report

This is an excellent paper and inclusion to the literature surrounding athlete-coach communication, particularly with respect to the menstrual cycle. I have very few edits to suggest – the authors are to be congratulated on an excellent and practical piece of research!

Specific comments below:

Abstract

This is a very well written abstract that concisely outlines the study background, aims, methods, and main results.

Introduction

This is a fantastic Introduction which includes up to date references and leads the reader clearly towards the study Methods

Line 42: Change emphatic to empathic/empathetic

Line 47: Be careful with the interpretation of the Larsen et al (2020) study findings – while HC user and individual athletes had statistically higher knowledge scores, the magnitude of this difference was low and the takeaway message was that almost all athletes had poor knowledge surrounding the MC/HC.

Methods

Table 1 and 2 should be placed after the Online Survey section as you haven’t yet explained the terms Commitment, Closeness and Complementarity which appear in the tables but are not explained in the table footnotes

Were data surrounding menstrual cycle status/hormonal contraceptive use specifically collected? It would be interesting to see if those with MC issues for example were more likely to communicate with their coach about these topics. This info could help provide insight into the enablers to communication between coaches and athletes

Results

It may be neater to read if the athlete and coach quotes are presented in a table?

Discussion

Lines 486-493: This section feels a bit speculative considering you did not report athletes’ MC/HC history, symptoms etc. except by way of athletes at times mentioning it throughout the discussions.

Otherwise, an excellent discussion of the results in the context of the broader research!

Author Response

General author response: Thank you for your encouraging feedback and useful suggestions for developing our article. Please find our responses to your specific comments outlined below.

Abstract

This is a very well written abstract that concisely outlines the study background, aims, methods, and main results.

Author response: Thank you, we appreciate this positive comment.

Introduction

This is a fantastic Introduction which includes up to date references and leads the reader clearly towards the study Methods.

Author response: Again, thank you for this encouraging compliment.

Line 42: Change emphatic to empathic/empathetic

Author response: Changed (L43), thank you for highlighting this typo.

Line 47: Be careful with the interpretation of the Larsen et al (2020) study findings – while HC user and individual athletes had statistically higher knowledge scores, the magnitude of this difference was low and the takeaway message was that almost all athletes had poor knowledge surrounding the MC/HC.

Author response: Thanks for highlighting this point. We’ve adapted this sentence to try to clarify the meaning (L74-78).

Methods

Table 1 and 2 should be placed after the Online Survey section as you haven’t yet explained the terms Commitment, Closeness and Complementarity which appear in the tables but are not explained in the table footnotes.

Author response: Thank you for this suggestion. We were unsure where to place the tables prior to submission, so this is a useful comment. We have now moved Tables 1 and 2 (although this change has not been “tracked” because the formatting became very messy).

Were data surrounding menstrual cycle status/hormonal contraceptive use specifically collected? It would be interesting to see if those with MC issues for example were more likely to communicate with their coach about these topics. This info could help provide insight into the enablers to communication between coaches and athletes.

Author response: This is another good point. Unfortunately we did not collect this information with the studied cohort. However, we are currently preparing a follow-up study with younger athletes and their coaches, where we will include questions about MC status and use of HCs in the questionnaire, so we will be able to incorporate these kind of analyses – thank you for the suggestions!

Results

It may be neater to read if the athlete and coach quotes are presented in a table?

Author response: We appreciate this suggestion and have considered once again how best to present our data. We would prefer to keep the quotes within the main text in order to provide context around each quote and to maintain the flow of the text. We hope this reads ok.

Discussion

Lines 486-493: This section feels a bit speculative considering you did not report athletes’ MC/HC history, symptoms etc. except by way of athletes at times mentioning it throughout the discussions.

Author response: This is true, thank you for the comment. We have now re-structured the sentences (L488-491) to clarify our findings, in relation to previous work. Again, we plan to collect this type of questionnaire data during our next study with junior athletes.

Otherwise, an excellent discussion of the results in the context of the broader research!

Author response: Once again, thank you for your kind words and encouraging feedback.

Reviewer 2 Report

The article seems very interesting to me with a great job of writing and bibliographic search. I believe that it is research that must be taken into account and that confirms that there are still taboo topics in 2020 in terms of the menstrual cycle and contraceptive hormonal treatment. That is why she considers this article to be relevant, and that it shows the current situation of a very important issue for elite women's sport and it is a question that must be addressed, and that it serves as an example for the scientific community.

Author Response

Author response: Thank you for your positive response to our work!

Reviewer 3 Report

Generally an excellent, well-reasoned paper.

One quick point. In the discussion, it might also be important to think about the structure of elite sport that creates the need for lack of MC.  That is, you talk about elite sport as a structural barrier, are there alternative practices or ways of approaching this?

Author Response

Author response: Thank you for your encouraging feedback and this very thoughtful comment, it’s a really interesting view of “elite sport”. Indeed, it could be relevant to consider whether cross-country skiing competition tracks could (should?) be adapted to suit a wider range of athlete types: e.g., more powerful skiers with greater sprinting abilities versus smaller/lighter skiers who are better in uphill terrain. We feel that this is somewhat beyond the scope of our research question, but we have included some additional sentences at L567-572. We hope this addition doesn’t appear “out of place”.